# Harmonization of Human Biomonitoring Studies in Europe: Characteristics of the HBM4EU-Aligned Studies Participants

**DOI:** 10.3390/ijerph19116787

**Published:** 2022-06-01

**Authors:** Liese Gilles, Eva Govarts, Laura Rodriguez Martin, Anna-Maria Andersson, Brice M. R. Appenzeller, Fabio Barbone, Argelia Castaño, Dries Coertjens, Elly Den Hond, Vazha Dzhedzheia, Ivan Eržen, Marta Esteban López, Lucia Fábelová, Clémence Fillol, Carmen Franken, Hanne Frederiksen, Catherine Gabriel, Line Småstuen Haug, Milena Horvat, Thórhallur Ingi Halldórsson, Beata Janasik, Nataša Janev Holcer, Réka Kakucs, Spyros Karakitsios, Andromachi Katsonouri, Jana Klánová, Tina Kold-Jensen, Marike Kolossa-Gehring, Corina Konstantinou, Jani Koponen, Sanna Lignell, Anna Karin Lindroos, Konstantinos C. Makris, Darja Mazej, Bert Morrens, Ľubica Palkovičová Murínová, Sónia Namorado, Susana Pedraza-Diaz, Jasmin Peisker, Nicole Probst-Hensch, Loïc Rambaud, Valentina Rosolen, Enrico Rucic, Maria Rüther, Dimosthenis Sarigiannis, Janja Snoj Tratnik, Arnout Standaert, Lorraine Stewart, Tamás Szigeti, Cathrine Thomsen, Hanna Tolonen, Ása Eiríksdóttir, An Van Nieuwenhuyse, Veerle J. Verheyen, Jelle Vlaanderen, Nina Vogel, Wojciech Wasowicz, Till Weber, Jan-Paul Zock, Ovnair Sepai, Greet Schoeters

**Affiliations:** 1VITO Health, Flemish Institute for Technological Research (VITO), 2400 Mol, Belgium; eva.govarts@vito.be (E.G.); laura.rodriguezmartin@vito.be (L.R.M.); arnout.standaert@vito.be (A.S.); veerle.verheyen@vito.be (V.J.V.); greet.schoeters@vito.be (G.S.); 2Department of Growth and Reproduction, Copenhagen University Hospital-Rigshospitalet, 2100 Copenhagen, Denmark; anna-maria.andersson@regionh.dk (A.-M.A.); hanne.frederiksen@regionh.dk (H.F.); 3Department of Precision Health, Luxembourg Institute of Health, 1445 Strassen, Luxembourg; brice.appenzeller@lih.lu; 4Department of Medicine—DAME, University of Udine, Via Colugna 50, 33100 Udine, Italy; fabio.barbone@uniud.it; 5Centro Nacional de Sanidad Ambiental (CNSA), Instituto de Salud Carlos III, 28029 Madrid, Spain; castano@isciii.es (A.C.); m.esteban@isciii.es (M.E.L.); spedraza@isciii.es (S.P.-D.); 6Department of Sociology, University of Antwerp, 2020 Antwerp, Belgium; dries.coertjens@uantwerpen.be (D.C.); bert.morrens@uantwerpen.be (B.M.); 7Provincial Institute for Hygiene, 2000 Antwerp, Belgium; elly.denhond@provincieantwerpen.be (E.D.H.); carmen.franken@provincieantwerpen.be (C.F.); 8Environmental Engineering Laboratory, Department of Chemical Engineering, Aristotle University of Thessaloniki, 54124 Thessaloniki, Greece; vazhajejeya@gmail.com (V.D.); katerinagabriel79@gmail.com (C.G.); spyrosk@auth.gr (S.K.); denis@eng.auth.gr (D.S.); 9HERACLES Research Center on the Exposome and Health, Center for Interdisciplinary Research and Innovation, Balkan Center, Bldg. B, 10th km Thessaloniki-Thermi Road, 57001 Thessaloniki, Greece; 10National Institute of Public Health, 1000 Ljubljana, Slovenia; ivan.erzen@nijz.si; 11Faculty of Public Health, Slovak Medical University, 833 03 Bratislava, Slovakia; lucia.fabelova@szu.sk (L.F.); lubica.murinova@szu.sk (Ľ.P.M.); 12Santé Publique France, Environmental and Occupational Health Division, 94415 Saint-Maurice, France; clemence.fillol@santepubliquefrance.fr (C.F.); loic.rambaud@santepubliquefrance.fr (L.R.); 13Division for Climate and Environmental Health, Norwegian Institute of Public Health, 0213 Oslo, Norway; linesmastuen.haug@fhi.no (L.S.H.); cathrine.thomsen@fhi.no (C.T.); 14Department of Environmental Sciences, Jožef Stefan Institute, 1000 Ljubljana, Slovenia; milena.horvat@ijs.si (M.H.); darja.mazej@ijs.si (D.M.); janja.tratnik@ijs.si (J.S.T.); 15Faculty of Food Science and Nutrition, University of Iceland, 102 Reykjavik, Iceland; tih@hi.is (T.I.H.); asav@hi.is (Á.E.); 16Nofer Institute of Occupational Medicine (NIOM), 91-348 Lodz, Poland; beatajan@imp.lodz.pl (B.J.); wojciech@imp.lodz.pl (W.W.); 17Croatian Institute of Public Health, Division for Environmental Health, 1000 Zagreb, Croatia; natasa.janev@hzjz.hr; 18Department of Social Medicine and Epidemiology, Faculty of Medicine, University of Rijeka, 51000 Rijeka, Croatia; 19National Public Health Center, 1097 Budapest, Hungary; kakucs.reka@nnk.gov.hu (R.K.); szigeti.tamas@nnk.gov.hu (T.S.); 20Cyprus State General Laboratory, Ministry of Health , P.O. Box 28648, 2081 Nicosia, Cyprus; akatsonouri@sgl.moh.gov.cy; 21Masaryk University Research Centre for Toxic Compounds in the Environment (RECETOX), 625 00 Bohunice, Czech Republic; jana.klanova@recetox.muni.cz; 22Department of Clinical Pharmacology, Pharmacy and Environmental Medicine, University of Southern Denmark, 5000 Odense, Denmark; tkjensen@health.sdu.dk; 23German Environment Agency (UBA), 14195 Berlin, Germany; marike.kolossa@uba.de (M.K.-G.); jasmin.peisker@uba.de (J.P.); enrico.rucic@uba.de (E.R.); maria.ruether@googlemail.com (M.R.); nina.vogel@uba.de (N.V.); till.weber@uba.de (T.W.); 24Cyprus International Institute for Environmental and Public Health, Cyprus University of Technology, 3603 Limassol, Cyprus; corina.konstantinou@cut.ac.cy (C.K.); konstantinos.makris@cut.ac.cy (K.C.M.); 25Department of Public Health and Welfare, Finnish Institute for Health and Welfare (THL), P.O. Box 30, 00271 Helsinki, Finland; jani.koponen@thl.fi; 26Swedish Food Agency, 751 26 Uppsala, Sweden; sanna.lignell@slv.se (S.L.); annakarin.lindroos@slv.se (A.K.L.); 27National Institute of Health, 1649-016 Lisbon, Portugal; sonia.namorado@insa.min-saude.pt; 28Public Health Research Centre, NOVA National School of Public Health, Universidade NOVA de Lisboa, 1099-085 Lisbon, Portugal; 29Department of Epidemiology and Public Health, Swiss Tropical and Public Health Institute, 4051 Basel, Switzerland; nicole.probst@unibas.ch; 30Department of Clinical Research, University of Basel, 4051 Basel, Switzerland; 31Institute for Maternal and Child Health—IRCCS “Burlo Garofolo”, 34137 Trieste, Italy; valentina.rosolen@burlo.trieste.it; 32Environmental Health Engineering, Institute of Advanced Study, Palazzo del Broletto—Piazza della Vittoria 15, 27100 Pavia, Italy; 33UK Health Security Agency, London SE1 8UG, UK; lorraine.stewart@phe.gov.uk (L.S.); ovnair.sepai@phe.gov.uk (O.S.); 34Department of Health Security, Finnish Institute for Health and Welfare (THL), P.O. Box 95, 70701 Kuopio, Finland; hanna.tolonen@thl.fi; 35Department Health Protection, Laboratoire National de Santé, 3555 Dudelange, Luxembourg; an.vannieuwenhuyse@lns.etat.lu; 36Department of Biomedical Sciences, University of Antwerp, 2020 Antwerp, Belgium; 37Institute for Risk Assessment Sciences (IRAS), Utrecht University, 3508 TC Utrecht, The Netherlands; j.j.vlaanderen@uu.nl; 38National Institute for Public Health and the Environment (RIVM), 3721 MA Bilthoven, The Netherlands; jan-paul.zock@rivm.nl

**Keywords:** human biomonitoring, joint HBM4EU survey, children, teenagers, adults

## Abstract

Human biomonitoring has become a pivotal tool for supporting chemicals’ policies. It provides information on real-life human exposures and is increasingly used to prioritize chemicals of health concern and to evaluate the success of chemical policies. Europe has launched the ambitious REACH program in 2007 to improve the protection of human health and the environment. In October 2020 the EU commission published its new chemicals strategy for sustainability towards a toxic-free environment. The European Parliament called upon the commission to collect human biomonitoring data to support chemical’s risk assessment and risk management. This manuscript describes the organization of the first HBM4EU-aligned studies that obtain comparable human biomonitoring (HBM) data of European citizens to monitor their internal exposure to environmental chemicals. The HBM4EU-aligned studies build on existing HBM capacity in Europe by aligning national or regional HBM studies. The HBM4EU-aligned studies focus on three age groups: children, teenagers, and adults. The participants are recruited between 2014 and 2021 in 11 to 12 primary sampling units that are geographically distributed across Europe. Urine samples are collected in all age groups, and blood samples are collected in children and teenagers. Auxiliary information on socio-demographics, lifestyle, health status, environment, and diet is collected using questionnaires. In total, biological samples from 3137 children aged 6–12 years are collected for the analysis of biomarkers for phthalates, HEXAMOLL^®^ DINCH, and flame retardants. Samples from 2950 teenagers aged 12–18 years are collected for the analysis of biomarkers for phthalates, Hexamoll^®^ DINCH, and per- and polyfluoroalkyl substances (PFASs), and samples from 3522 adults aged 20–39 years are collected for the analysis of cadmium, bisphenols, and metabolites of polyaromatic hydrocarbons (PAHs). The children’s group consists of 50.4% boys and 49.5% girls, of which 44.1% live in cities, 29.0% live in towns/suburbs, and 26.8% live in rural areas. The teenagers’ group includes 50.6% girls and 49.4% boys, with 37.7% of residents in cities, 31.2% in towns/suburbs, and 30.2% in rural areas. The adult group consists of 52.6% women and 47.4% men, 71.9% live in cities, 14.2% in towns/suburbs, and only 13.4% live in rural areas. The study population approaches the characteristics of the general European population based on age-matched EUROSTAT EU-28, 2017 data; however, individuals who obtained no to lower educational level (ISCED 0–2) are underrepresented. The data on internal human exposure to priority chemicals from this unique cohort will provide a baseline for Europe’s strategy towards a non-toxic environment and challenges and recommendations to improve the sampling frame for future EU-wide HBM surveys are discussed.

## 1. Introduction

The presence of chemical pollutants in our environment is ubiquitous and part of our modern way of life [1]. Some of these chemicals are hazardous and pose risks to human health. To assess the potential health risks, a good understanding of the extent of human exposure is needed, together with the toxic potency of the chemicals. It is often difficult to predict individual exposure based on environmental exposure measurements (e.g., water, food, soil), given that they are usually multi-pathway in nature [2]. Human biomonitoring (HBM) is an important tool to measure the concentration of chemicals present in the human body. It reflects aggregate exposure via different pathways such as through our living or work environment, our diet, or the use of consumer products. In the US and Canada, human exposure to chemicals is monitored through biennial national human biomonitoring campaigns. These HBM surveys were already initiated in 1999 (National Health and Nutrition Examination Survey (NHANES, CDC, Atlanta, GA, USA) and 2007 (Canadian Health Measures Survey (CHMS, Statistics Canada, Ottawa, Canada). Also in Korea in 2005, the first National Survey for Environmental Pollutants in the Human Body (KorSEP) was conducted followed by the Korean National Environmental Health Survey (KoNEHS) [3] which was implemented in 2009 as a recurrent program with a 3-year interval period. More recently, in 2017, also China initiated an HBM program called CHBM (China National Human Biomonitoring) [2]. In Europe, a coherent European recurrent HBM surveillance program is lacking.

In 2017 the European Human Biomonitoring Initiative (HBM4EU) was launched with the ambition to coordinate and advance human biomonitoring in Europe. The project is a joint effort of 30 European countries, and the European Environment Agency, co-funded by the European Commission under Horizon 2020 [4]. As part of the HBM4EU project, the HBM4EU-aligned studies were set up to collect harmonized and quality controlled recent internal exposure data from European citizens to environmental pollutants. The aim of these HBM4EU-aligned studies is to collect a population sample for children (6–11 years), teenagers (12–19 years), and adults (20–39 years) that is representative of sex (males, females) and includes individuals from cities, towns/suburbs, and rural areas, with different socio-economic status (SES), and recruited from the four geographical regions of Europe (north, east, south, and west) as defined by the UN geoscheme.

Since some European countries already have a recurrent HBM program, i.e., Germany [5], Belgium (Flemish region) [6], France [7], Czech Republic [8], Sweden [9], and Slovenia [10,11] it was decided to make use of this existing capacity/infrastructure instead of setting up a completely new European coordinated study as was conducted previously during the COPHES/DEMOCOPHES project [12]. The HBM4EU-aligned studies have brought together ongoing and newly planned HBM initiatives in Europe and harmonized them. The sampling scheme, which describes the criteria for HBM studies in Europe to be included in this harmonization exercise, resulting in the HBM4EU-aligned studies, is published elsewhere [13].

In HBM4EU, a total of 18 chemical substance groups were prioritized over two rounds [14]. In the HBM4EU-aligned studies, exposure to a reduced selection of these prioritized chemicals has been investigated, i.e., phthalates and Hexamoll^®^ DINCH, flame retardants, per- and polyfluoroalkyl substances (PFASs), polycyclic aromatic hydrocarbons (PAHs), bisphenols and cadmium (from the first list of priority substances) and pesticides, arsenic species, UV filters (benzophenones), mycotoxins and acrylamide (from the second list of priority substances). The resulting HBM data indicate the present chemical exposure of European citizens (spatial trends). It can be used to identify (sub)populations at risk and highlight inequalities, to compare with exposure levels measured in internationally established HBM campaigns such as NHANES [15], CHMS [16], KNHANES [17], JECS [18], CNHBM [2] or previous European HBM projects such as COPHES/DEMOCOPHES [19] and to identify variables that may influence internal exposure levels. The HBM data can also be input into models to estimate external intakes or the dose to target organs. Furthermore, connecting HBM data to biomarkers of early effects may increase knowledge of the underlying processes that could eventually lead to adverse health outcomes [20]. All this information supports policymakers in designing and evaluating measures or regulations such as REACH to protect the population against adverse effects of environmental pollutants. This paper provides a transparent description of the characteristics of the studies and the population sample in which exposure to the first set of priority substances was measured. It compares the obtained population sample with the general European population for the following characteristics: sex, residential degree of urbanization, and educational level of the household, based on EUROSTAT data (see Section 3 results). It reflects on the adaptations made from the theoretical sampling frame described earlier by Gilles et al. to accommodate the inclusion of sufficient studies to obtain comparable HBM data of exposure to environmental pollutants, prioritized under HBM4EU, with European wide coverage. We discuss the strengths and limitations of using the newly generated HBM data to inform environment and health policies [4].

## 2. Materials and Methods

The aim of the HBM4EU-aligned studies is to collect comparable internal exposure data for priority substances in a sample of the European population that is representative of sex (males, females) and includes individuals from cities, towns/suburbs, and rural areas, with different socio-economic status (SES), and recruited from the four geographical regions of Europe (north, east, south, and west) as defined by the UN geoscheme. The population sample was selected by utilizing a sampling scheme designed to guide the inclusion of eligible studies. The scheme is based on the inclusion of 11 to 12 primary sampling units (PSUs) per age group (children 6–11 years, teenagers 12–19 years, and adults 20–39 years), distributed proportionally to the number of inhabitants in the four geographical regions (north, east, south, west) of Europe, with a maximum contribution of 300 participants per PSU. The PSUs are HBM studies in European countries that fulfilled the preset inclusion criteria: (i) completed studies with available biobank samples, (ii) studies that were initiated before the start of the HBM4EU project but with sampling within the stipulated timeframe, (iii) new studies, adopting the HBM4EU protocols. Furthermore, samples had to be collected between 2014 and 2020, fall within the aforementioned age groups, analysis had to be performed in laboratories that successfully participated in the HBM4EU QA/QC program [21], and the data must be shared on the EU level. The sampling scheme was described in more detail by Gilles et al. [13].

Results will be stratified according to sex, residence’s degree of urbanization, and educational level of the household. Therefore, additional criteria were set for PSUs to contribute with a 50:50 ratio of male and female participants, with individuals living in rural areas, in towns/suburbs, and in cities, and with at least 10% of individuals from low, medium, and high educational level. The residential area of participants (cities, towns and suburbs, rural areas) was characterized by a high, medium, and low degree of urbanization according to the DEGURBA classification from EUROSTAT (2018) [13]. Educational level was used as a surrogate for SES. The classification was based on the International Standard Classification of Education (ISCED) developed by the United Nations Educational, Scientific and Cultural Organization (UNESCO). We distinguished three levels: no to lower secondary education (ISCED level 0–2), attaining upper secondary to post-secondary non-tertiary education (ISCED level 3–4), and attaining tertiary education or higher (ISCED level ≥5). Children and teenagers were categorized according to the households’ educational attainment [14].

### 2.1. Studies Included in the HBM4EU Aligned Studies

The following section includes an overview table of the studies that were selected as PSUs and participated in the HBM4EU-aligned studies. In total, 25 studies from 21 countries contributed to the HBM4EU-aligned studies. Some studies provided samples for more than one age group, e.g., ESTEBAN, GerES V, SLO CRP, CROME, and NEB II. Furthermore, some of the studies had a larger sample size than the maximal contribution of 300 samples; in those cases, a selection of 300 individuals was performed following a step-wise selection procedure. In short, the total study population was stratified into mutually exclusive subgroups followed by a random selection of the participants adhering to the subgroup proportions defined (i.e., sex of the participant (50% of each sex should be represented), degree of urbanization (at least 10% of each of the three levels should be represented), educational level (at least 10% of each of the three levels should be represented), sampling season (25% of each season), age (all available ages that fall within the categories 6–11, 12–19 or 20–39 must be covered).

Contributing studies are listed alphabetically in Table 1 and a short description with additional information per study is provided in the Appendix A.

The data from the PSUs are pooled into a European sample of children, teenagers, and adults. The baseline characteristics of the sampled population of each age group (children, teenagers, and adults) are described per study (PSU) (Appendix A), per European region (north, east, south, west), and for the total sampled population (Table 1, Table 2 and Table 3). Verification of the contributing studies was undertaken to ascertain whether they met the predefined criteria for representativeness: sex, attained educational level (ISCED), and degree of urbanization (DEGURBA). Furthermore, the characteristics of the HBM4EU-aligned studies sample of each age group were compared to those of the European general population based on EUROSTAT’s tables for EU-28, 2017 (see Section 3).

### 2.2. Biological Specimens Collected

An overview of the biological specimens collected and used within the scope of the HBM4EU-aligned studies is presented in Appendix A. Urine samples were collected in all three age groups. Different urinary sample types (i.e., first morning void, random spot samples, and 24 h urine samples) were obtained from the study participants. Children provided first morning voids in six PSUs and random spot samples in six PSUs. Six PSUs collected first morning voids from teenagers and five PSUs collected random spot samples. In adults, first morning voids were collected in five PSUs, five PSUs collected random spot samples and one PSU collected 24 h samples. All PSUs had to report urinary creatinine (crt) concentrations. Specific gravity was also measured in urine samples from teenagers (except in ESTEBAN). In addition to urine samples, blood samples were collected from children and teenagers for the analysis of brominated flame retardants (BFRs) and PFAS, respectively. Blood serum is the preferred matrix for both exposure measures [40]. Only five PSUs of children (NEB II, SLO CRP, CROME, ESTEBAN, and 3 × G) had blood samples available for analysis of chemical exposure within the frame of the HBM4EU-aligned studies. Four of these five PSUs provided serum samples, one PSU provided plasma samples for analysis. Nine teenagers’ PSUs (NEB II, Riksmaten Adolescents 2016–2017, PCB cohort (follow-up), SLO CRP, CROME, BEA, ESTEBAN, FLEHS IV, and GerES V-sub (unweighted)) collected blood samples: seven PSUs collected serum and two collected plasma. Most of those PSUs collected non-fasting samples (9 non-fasting vs. 5 fasting). To standardize concentration levels of lipid-soluble chemicals such as the BFRs, total lipid content was calculated in all PSUs using an enzymatic summation method. However, the exact formula used was different for each of the five PSUs, that analyzed BFRs. To further improve comparability, the total blood lipid content was recalculated for all PSUs using the formula: Total lipids (mg/dL) = 2.27 ∗ (Total CHOL) + TRIGL + 62.3 mg/dL proposed by Bernert et al. [41].

### 2.3. Exposure Assessment

A detailed overview of the individual studies and the number of participants with biomarker data available for a specific priority substance is presented in Table 2. These numbers differ per substance group. As 50% of the analytical costs had to be co-financed by the studies themselves, the studies were not obliged to measure each prioritized group of substances. As a result, data on exposure to phthalates and HEXAMOLL^®^ DINCH is available for 2880 children and 2799 teenagers. Data on BFRs and OPFRs are available for 711 and 1770 children respectively. Data on PFASs exposure are available for 1957 teenagers and data on biomarkers for bisphenols, Cd and PAHs are available for 2756, 2510, and 2609 adults, respectively.

**Table 2 ijerph-19-06787-t002:** Number of subjects analyzed per substance group for the entire HBM4EU-aligned studies and per study.

Study	Country	Phthalates and HEXAMOLL^®^ DINCH	BFRs	OPFRs	PFASs	Cd	Bisphenols	PAH
**Children**								
NEBII	NO	300	300	300	-	-	-	-
OCC	DK	300	0	291	-	-	-	-
InAirQ	HU	262	0	0	-	-	-	-
PCB cohort	SK	300	0	300	-	-	-	-
POLAES	PL	300	0	0	-	-	-	-
SLO CRP	SL	149	130	147	-	-	-	-
CROME	EL	161	55	0	-	-	-	-
NAC II	IT	300	0	0	-	-	-	-
ESTEBAN	FR	286	226	299	-	-	-	-
GerES V-sub(unweighted)	DE	300	0	300	-	-	-	-
3 × G	BE	133	0	133	-	-	-	-
SPECIMEn-NL	NL	89	0	0	-	-	-	-
Total		**2880**	**711**	**1770**	-	-	-	-
**Teenagers**								
NEB II	NO	181	-	-	177	-	-	-
Riksmaten Adolescents 2016–2017	SE	300	-	-	300	-	-	-
POLAES	PL	281	-	-	0	-	-	-
CELSPAC: TE	CZ	300	-	-	0	-	-	-
PCB cohort (follow-up)	SK	287	-	-	292	-	-	-
SLO CRP	SL	96	-	-	94	-	-	-
CROME	EL	150	-	-	52	-	-	-
BEA	ES	300	-	-	299	-	-	-
ESTEBAN	FR	304	-	-	143	-	-	-
FLEHS IV	BE	300	-	-	300	-	-	-
GerES V-sub(unweighted)	DE	300	-	-	300	-	-	-
Total		**2799**	-	-	**1957**	-	-	-
**Adults**								
Diet_HBM	IS	-	-	-	-	203	203	203
FinHealth	FI	-	-	-	-	0	300	0
POLAES	PL	-	-	-	-	228	228	228
(C)ELSPAC: YA	CZ	-	-	-	-	300	290	300
HBM survey in adults in Croatia	HR	-	-	-	-	300	300	300
INSEF-ExpoQuim	PT	-	-	-	-	295	296	296
ESTEBAN	FR	-	-	-	-	393	163	201
HBM4EU study for Switzerland	CH	-	-	-	-	0	300	300
ESB	DE	-	-	-	-	289	180	331
Oriscav-Lux2	LU	-	-	-	-	210	209	210
CPHMINIPUB (parents)/DYMS	DK	-	-	-	-	292	287	240
**Total**		-	-	-	-	**2510**	**2756**	**2609**

Phthalates and DINCH, OPFRs, Cd, bisphenols and PAHs are analyzed in urine samples, BFRs are analyzed in blood.

In total exposure to the following 60 exposure biomarkers was assessed: 15 phthalate metabolites: MEP, MBzP, MiBP, MnBP, MCHP, MnPeP, MEHP, 5OH-MEHP, 5oxo-MEHP, 5cx-MEPP, MnOP, OH-MiNP, cx-MiNP, OH-MiDP, cx-MiDP; 2 HEXAMOLL^®^ DINCH metabolites: OH-MINCH and cx-MINCH; 10 BFRs: TBBPA, DBDPE, 2,4,6-TBP, BDE-47, BDE-153, BDE-209, DP-syn, DP-anti, α-HBCD, γ-HBCD, and 4 OPFRs: DPHP, BDCIPP, BCEP, and BCIPP; 12 PFASs: PFPeA, PFHxA, PFHpA, PFOA, PFNA, PFDA, PFUnDA, PFDoDA, PFBS, PFHxS, PFHpS and PFOS (sum of all isomers); 3 bisphenol substances: BPA and substitutes BPS and BPF; 13 PAHs: 1-naphthol, 2-naphthol, 1,2 DHN, 2-FLUO, 3-FLUO, 9-FLUO, 1-PHEN, 2-PHEN, 3-PHEN, 4-PHEN, 9-PHEN, 1-PYR, 3-BaP, and cadmium. To safeguard the comparability of the data, analysis had to be performed in laboratories that successfully participated in the HBM4EU QA/QC program [21]. The laboratories could choose for which analytes they participated in the QA/QC program and the analytical qualification was evaluated at the metabolite level [42,43]. As a result, the biomarkers for which data are available differ somewhat per PSU. The biomarker data for each of the data collections have a data quality label assigned. A detailed overview of analytes measured per study is available in Appendix A. The detection limit (LOD) and quantification limit (LOQ) reported vary among the studies, as well as the methods used for the determination of LOD/LOQ. The LOQ is used as a cut-off to report quantifiable data, except when a lab only reported a LOD, then the LOD was used as a cut-off to report quantifiable data. Values below the cut-off are imputed.

## 3. Results

The HBM4EU-aligned studies build on existing capacity in Europe by aligning HBM studies in Europe. As a result of this approach, studies with different designs and degrees of representativeness were combined. In total 9609 European citizens (3137 children, 2950 teenagers, and 3522 adults) were sampled between 2014 and 2021. Twenty-five studies contributed to the HBM4EU-aligned studies. Sixty-eight percent of the studies were initiated or conducted before HBM4EU and biobanked samples were made available to the project, 32% of the studies were initiated specifically for HBM4EU such as in Greece, Czech Republic, Iceland, Croatia, Portugal, and Switzerland and implemented the guideline protocols developed under HBM4EU [44]. The HBM4EU-aligned studies include participants from longitudinal cohort studies (OCC, NEB II, PCB cohort, NAC II, 3 × G, and CELSPAC:YA), and cross-sectional surveys (InAirQ, SLO CRP, CROME, CELSPAC:TE, BEA, FLEHS IV, HBM4EU study in Switzerland, HBM in adults in Croatia, ESB, and Oriscav-Lux2). Some of the studies were part of a national nutrition survey (Riksmaten Adolescents 16–17, Diet_HBM and ESTEBAN) or a national health survey (ESTEBAN, GerES V-sub (unweighted) and INSEF-ExpoQuim) that had an HBM module incorporated and two cross-sectional case-control studies (SPECIMEn-NL, POLAES) were included. Moreover, the representativeness differs: 11 studies were representative at the national level, and 14 studies at the regional level (see Table 1).

The participation rate strongly varied among the contributing studies ranging from 12.5% to 75.7%. Seventy percent of the studies offered a summary of the group results and a personal report with the participants’ individual results as an incentive, one study presented the results to participants on the group level only. In some European countries, e.g., Finland, the provision of monetary incentives is not allowed. Furthermore, it can be noticed that those studies with a higher participation rate provided additional incentives in the form of a small present (e.g., study mascot as a cuddly toy, key ring, fisheye lens for a mobile phone, shopping voucher, entrance ticket for a recreation domain, or a small amount of cash).

### 3.1. Geographical Distribution of the Sample

The children of the HBM4EU-aligned studies were recruited in 12 different PSUs (European countries). Teenagers and adults were recruited in 11 different European countries. Figure 1 illustrates the geographical distribution of the study populations based on EUROSTAT’s NUTS 2 (nomenclature of territorial units for statistics) classification. The children from ESTEBAN, GerES V sub (unweighted), and NEB II had a nationwide geographical distribution whereas the individuals from the other studies were recruited from a more limited geographical region. Teenagers from ESTEBAN, Riksmaten Adolescents, and NEB II contributed with a sample that has national geographical coverage, the other studies had a regional geographical coverage. More adult studies contributed with a national geographical representative sample: ESTEBAN, FinHealth, HBM in Croatia, HBM in Switzerland, Oriscav-Lux2, INSEF-ExpoQuim. The other five studies had regional geographical coverage. None of the studies were known hotspots for the priority chemicals assessed.

### 3.2. Baseline Characteristics of Study Participants

The baseline characteristics of each age group (children, teenagers, and adults) are described for the total sampled population Table 3, Table 4 and Table 5, per European region (north, east, south, west) and per study (PSU) (Appendix A). The total sampled population is compared to the “general European population” based on data from EUROSTAT EU-28, 2017, unless stated otherwise.

**Table 3 ijerph-19-06787-t003:** Characteristics of sampled population children by European region and in total.

Characteristics	Northern EU	Eastern EU	Southern EU	Western EU	EU Total	EU Reference
No. of participants	600	862	610	1065	3137	
Age (years)						
Median (p25–p75)	7 (7–10)	10 (9–11)	7 (7–9)	8 (7–10)	9 (7–10)	
Min-max	6–11	7–12	6 – 11	6–12	6–12	
Sampling period (year)						
Median (p25–p75)	2017.5 (2016–2019)	2017 (2016–2017)	2018 (2016–2020)	2015 (2015–2017)	2016 (2015–2018)	
Min-max	2016–2019	2014–2018	2014–2021	2014–2020	2014–2021	
Sex N (%)						
Girl	275 (45.8%)	448 (52.0%)	315 (51.6%)	514 (48.3%)	1552 (49.5%)	48.68%
Boy	325 (54.2%)	414 (48.0%)	295 (48.4%)	548 (51.5%)	1582 (50.4%)	51.32%
Missing	0 (0%)	0 (0%)	0 (0%)	3 (0.3%)	3 (0.1%)	
Residential degree of urbanization N (%)						
Cities	386 (64.3%)	464 (53.8%)	318 (52.1%)	215 (20.2%)	1383 (44.1%)	41.70%
Towns/suburbs	134 (22.3%)	221 (25.6%)	80 (13.1%)	476 (44.7%)	911 (29.0%)	31.00%
Rural area	80 (13.3%)	176 (20.4%)	212 (34.8%)	374 (35.1%)	842 (26.8%)	27.30%
Missing	0 (0%)	1 (0.1%)	0 (0%)	0 (0%)	1 (0.03%)	
Educational level of the household N (%)						
ISCED 0–2	41 (6.8%)	19 (2.2%)	36 (5.9%)	34 (3.2%)	130 (4.1%)	26.00%
ISCED 3–4	168 (28.0%)	393 (45.6%)	200 (32.8%)	342 (32.1%)	1103 (35.2%)	46.10%
ISCED ≥ 5	378 (63.0%)	411 (47.7%)	367 (60.2%)	676 (63.5%)	1832 (58.4%)	27.90%
Missing	13 (2.2%)	39 (4.5%)	7 (1.1%)	13 (1.2%)	72 (2.3%)	

EU reference population based on EUROSTAT tables: residential degree of urbanization based on EU-28, 2017; educational level of the household based on EU-28, 2017, age 15–64 years.; sex based on EU-28, 2017, age 5–9 years.

The HBM4EU sampled population of children has slightly more boys (50.4%) than girls (49.5%), which reflects the European population based on EUROSTAT 2017 data (boys: 51.3% vs. girls: 48.7%) (Table 3). The HBM4EU sampled population of teenagers has slightly more girls (50.6%) than boys (49.4%). This differs slightly from the European population (girls: 48.4%% vs. boys: 51.6%) (Table 4). The same is true for the HBM4EU adult population which consists of 52.6% women and of 47.4% men, whereas the European population has 49.4% women and 50.7% men (2017) (Table 5). In all age groups, the HBM4EU population sample approaches the 50:50 ratio that was targeted. At the level of the PSUs (studies), the 50:50 ratio of boys:girls, male:female is approached by all studies in all age groups with a maximum divergence of 31:69 ratio (POLAES adults, Appendix A).

The residence’s degree of urbanization, based on the DEGURBA classification by EUROSTAT [46], is similar in the HBM4EU children compared to Europe’s general population (41.7% cities, 31.0% towns/suburbs, 27.3% rural areas) (Table 3). Most children of the HBM4EU sample live in high population density areas i.e., cities (44.1%), 29.0% live in medium density areas (towns, suburbs) and 26.8% live in rural areas. Slightly more teenagers reside in cities (37.69%) than in towns/suburbs (31.2%) and rural areas (30.2%) (Table 4). Most adults of the HBM4EU-aligned studies live in cities (71.9%), 14.2% live in medium density areas (towns, suburbs) and 13.4% live in rural areas (Table 5). Compared to the European population of the same age group (20–39 years) (41.8% cities, 38.2% towns/suburbs, 20.0% rural; EU-28, 2017, 20–39 years), the sampled adult population from cities is overrepresented and adults from towns/suburbs and rural areas are underrepresented. If the children’s group is considered, only 42% of the PSUs reach the target of a minimum representation of 10% for each level of urbanization, and in 4 PSUs all children are recruited from a single degree of urbanization. For both teenagers and adults, 63% of the studies reach the minimum target of 10% of each level, in 2 PSUs all teenagers are from a single level, i.e., POLAES 100% cities and SLO CRP 100% rural areas, in 2 PSUs all adults are living in cities only.

The educational level of the household of children and teenagers was compared to the educational level of the European population from 15–64 years as reported by EUROSTAT (EU-28 population in 2017). Moreover, 26.0% of Europeans of that age group attained no to lower secondary education, 46.1% upper secondary to post-secondary non-tertiary education, and 27.9% tertiary education or higher. Most of the sampled children and teenagers live in households with one of the parents attaining tertiary education or higher (ISCED level ≥5 for 58.4% of the children and 54.5% of the teenagers) (Table 3 and Table 4). Only 4.1% of children and 6.1% of teenagers are from households with no to lower secondary education (ISCED level 0–2). Also in the adult sampled population, most participants (68.38%) attained tertiary education or higher (ISCED level ≥5), 24.4% attained upper secondary to post-secondary non-tertiary education (ISCED level 3–4), and only 6.7% no to lower secondary education (ISCED level 0–2) (Table 5). In each age group, individuals with no to lower secondary education are underrepresented. Considering the educational level of the household in each PSU separately: in 4 PSUs (NEB II, POLAES, CROME, SPECIMEN-NL) no children are sampled from households with ISCED level (0–2), 2 PSUs (POLAES, ESB) have no teenagers from households with no to lower secondary education. The percentage of smokers in the sampled adult population (17.6%) is somewhat lower compared to that of the general European adult population aged 18–44 years (22.9%) (Table 5).

It was preset that the children must be between 6 and 11 years of age. However, 3.9% of the children sampled are 12 years of age. PCB cohort, ESTEBAN, and GerES V-sub (unweighted) have 12 years old participants included (Appendix A). Some children of the PCB cohort were recruited at age 11 but had turned 12 at sampling. ESTEBAN selected 300 subjects from the original ESTEBAN study population based on the age of the subject at questionnaire conduct. As some time elapsed between questionnaire administration and sample collection, some individuals were 11 years old at the time the questionnaires were completed but had turned 12 years when the sample was provided. GerES V-sub (unweighted) collected blood and urine samples at different times. The selection of 300 subjects from the original GerES V survey was based on the age of the first sample (blood) collection. Consequently, some children were already 12 years when urine samples were collected. It was decided to accept these deviations and not exclude those individuals and their data from the HBM4EU-aligned studies. Depending on the research objective, it will be evaluated whether these subjects should be included or excluded from the dataset. Not all ages are equally represented in the HBM4EU children’s group (see Appendix A). Some studies recruited children over the entire age range from 6 to 11 years i.e., CROME whereas other studies include a narrower age range, e.g., NAC II, OCC (Appendix A). This leads to an overrepresentation of specific ages (see Figure 2). Most children (N = 978) are 7 years old (31.2%) while 6 years old children are underrepresented (4.4%). Also, the sampling dates varied among PSUs. For example: CROME children were sampled in 2020–2021, whereas the ESTEBAN children were sampled between 2014 and 2016. Most samples of the children’s age group were collected in 2015–2016-2017. Due to a delay in sample collection, in some of the PSUs, the sampling period was expanded from 2014–2020 to 2014–2021 for all three age groups.

The teenagers’ age group was predefined from 12 to 19 years; however, no 19-year-old individuals are present in the sample. Similar to the children group, not all ages are equally represented in the teenagers’ group. Some studies cover the entire age range from 12–18, whereas other studies such as FLEHS IV and BEA include a narrower age range (see Appendix A). This leads to an underrepresentation of specific ages (see Figure 2) with only fifteen teenagers of 18 years old. Additionally, the sampling dates varied between studies with most samples being collected in 2017.

The adults age group was predefined from 20 to 39 years of age because it provides a more homogeneous group with a more similar health status (reproductive age group) and it mirrors the age stratification used in NHANES and CHMS (CDC, 2017; Statistics Canada, 2018). The samples were collected between 2014 and 2021 with most samples collected in more recent years (2017–2021). Samples from 2014–2016 were only obtained in Western Europe while samples from Eastern Europe covered only 2017 and 2019 (Figure 2). Differences in specific ages and sampling years may influence biomarker concentrations and should be taken into account by multivariate statistical modeling when biomarker levels are compared between geographical regions. In addition, seasonal coverage varied among the PSUs: 50% of the studies collected samples across all 4 seasons, 17% of the studies cover 3 out of 4 seasons, 21% of the studies cover 2 seasons and 12% of studies are limited to collecting samples in just one season (see Appendix A). Especially for those exposures, with known seasonal variability (e.g., PAHs), samples are preferably collected all year round—covering all 4 seasons—to obtain representative exposure data.

### 3.3. Questionnaires

As part of the HBM4EU project, new harmonized questionnaires have been developed based on knowledge and experience in different EU countries (Pack et al. in preparation). As the HBM4EU-aligned studies combine ongoing and newly planned studies, the questionnaires used by the studies differ. Four studies, that were still in the planning phase, adopted the HBM4EU questionnaires or developed their own questionnaire based on the HBM4EU questionnaire (CROME, HBM4EU-study for Switzerland, Diet_HBM, and INSEF-ExpoQuim). The HBM4EU questionnaires are available online: https://doi.org/10.5281/zenodo.6414615. Questionnaires had different components and were differently administered (self-administered, telephone, in person, and computer assisted or on paper). In some studies, different collection methods were applied for separate questionnaire modules. An overview per study is available in Appendix A. Questions about socio-demographic characteristics were included in all studies, the majority of studies also captured information on dietary habits and health status of the participant (88%), lifestyle (85%), and residential environment (70%). Specific questionnaire components on exposure to specific sources of chemicals, use of consumer products, or information on recent exposure (e.g., last days before sampling), were less frequently included, 73%, 64%, and 50% respectively. Information on occupational exposure was collected in only 54% of adult studies. A closer inspection of the questionnaire component related to dietary habits reveals several differences: 58% have used a food frequency questionnaire (FFQ) asking about food consumption frequency and portion size and 17% have used a food propensity questionnaire (FPQ), asking about frequency of consumption only. Difficulty with these FPQ and FFQ are the differences in foods listed, length of the reference period, response intervals for specifying the frequency of use, and the procedure for estimating portion size. For example, some questionnaires asked for dietary habits over the last year, others over the last 3 months. In addition to FFQ and FPQ, some studies (26%) also requested information on recent dietary exposure through a 24 h recall questionnaire. This 24 h recall method was also recommended by EFSA for the harmonized collection of food consumption data in adults [47]. To harmonize the available data across the studies a post-harmonization approach was followed which is described elsewhere [13]. The final codebooks with harmonized variables are available online: HBM4EU Harmonized Codebook Adults-Aligned studies|Zenodo (https://doi.org/10.5281/zenodo.6598404); HBM4EU Harmonized Codebook Children-Aligned studies|Zenodo (https://doi.org/10.5281/zenodo.6598519); HBM4EU Harmonized Codebook Teenagers-Aligned studies|Zenodo (https://doi.org/10.5281/zenodo.6598532).

### 3.4. Data Accessibility

For each of the studies included in the HBM4EU-aligned studies, metadata are available in IPCHEM, the European Commission’s Information Platform for Chemical Monitoring. The summary statistics (percentiles P5, P10, P25, P50 P75 P90, P95) of the exposure biomarkers will be integrated into the openly accessible online European HBM dashboard (https://www.hbm4eu.eu/eu-hbm-dashboard/) and IPCHEM (https://ipchem.jrc.ec.europa.eu/), where it can be accessed. Data sharing of individual-level data is possible upon request.

## 4. Discussion

### 4.1. Strengths and Limitations

Overall, the adopted approach resulted in a sample of 3137 children (6–12 years), 2950 teenagers (12–18 years), and 3522 adults (20–39 years) which is comparable to the general European population (EUROSTAT EU-28, 2017). The HBM4EU-aligned studies provide a large coherent dataset that allows making robust conclusions regarding internal exposure to pollutants in European children, teenagers, and adults, as well as stratified by sex, educational level, degree of urbanization, and European region. Furthermore, the quality and comparability of analytical results are guaranteed through the extensive QA/QC scheme organized as part of HBM4EU [21]. Working with a network of multiple laboratories provided a large capacity and therefore samples could be analyzed in a relatively short timeframe. Some points that can be improved are lowering and harmonization of detection and quantification limits (LOD, LOQ) across laboratories and additional laboratory training to laboratories across Europe to analyze the same broad set of metabolites including newer (emerging) chemicals. This will in turn improve the potential for exploring co-occurrent internal exposures (i.e., personal mixtures). With the current dataset, this is already possible, but to a limited extent because the combination of available data varies per study (Appendix A).

The HBM4EU-aligned studies dataset combines information on socio-economic characteristics, health status, lifestyle, food habits, and residential environment, from the different studies into a large EU-wide and comprehensive dataset for the interpretation of internal exposure levels. The post-harmonization approach applied, is a time-consuming process and also presents some limitations. For example, information on the consumption of canned food items was available for only 4/11 studies in adults. Missing variables in some studies will reduce the sample size available for the analysis of exposure determinants. Upfront alignment of questionnaires in future surveys will increase the comparability and use of the European pooled dataset.

Pooling data from different studies increases the sample size and statistical power to study exposure–effect associations. For a European study that is only partly financed with EU funds, and which requires financial commitments from the member states, it is not feasible to impose a fully harmonized questionnaire. It was emphasized by the participating countries that flexibility with regard to questionnaires is necessary, on the one hand, to guarantee continuity with previous campaigns, for those countries that already have a recurrent HBM program, but on the other hand, also to include culturally and nationally bound information. As a feasible solution, a basic list of variables must be defined that can be retrieved in a harmonized way in all studies.

Another challenging obstacle is to enhance comparability of the exposure levels measured in different urine sample types (first morning, random spot, or 24 h sample). Exposure biomarkers in urine samples can be reported on a volume basis but can also be standardized for urinary dilution. Several methods are available; however, there is no consensus on the most appropriate method to account for the variations in dilution among spot urine samples [48]. The use of different methods across the suite of studies complicates the comparison of results [49]. Creatinine is most commonly used as a marker for urinary dilution and was measured in most HBM4EU urine samples allowing comparison with international and historical exposure data. However, as the creatinine (crt) concentration can be strongly influenced by growth, its suitability as a marker of urinary dilution is increasingly questioned. Therefore, an analysis of specific gravity (SG) was performed on teenagers’ samples. The inclusion of both SG and crt measurements in all age groups is supported by those parties involved in the next HBM program. When analyszing the data, urine sample type will be added as a variable in the statistical models. Additionally, the type of blood samples (serum or plasma) differed per study. However, it is not expected that the matrix type (serum vs. plasma) will impact the measured levels of PFASs and BFRs, as a 1:1 serum-to-plasma ratio for PFHxS, PFOS, and PFOA was previously demonstrated [50]. BFRs bind to lipids. They are measured in the lipid fraction extracted from serum or plasma which have similar blood lipid concentrations [51].

The HBM4EU-aligned studies dataset provides a good picture of the internal exposure of European citizens. However, there are some points that can be improved. Individuals with lower secondary education attainment are underrepresented in our population sample. This has been observed previously in epidemiological studies [52]. Future surveys should include additional efforts to attempt to engage individuals attaining up to lower secondary education (ISCED level 0–2). Furthermore, the final sampling period of the HBM4EU-aligned studies is fairly wide 2014–2021, spanning a time period of eight years. In addition, the time periods differ per PSU (Appendix A). Such a broad sampling period may introduce a time trend effect within the sample. For the next European HBM program, it is recommended to shorten the sampling period to a three-year period, to improve on this aspect. With regard to the age of the participants, there are also differences per PSU (Appendix A) resulting in an over- or underrepresentation of specific ages. For example, no teenagers of 19 years were included. Ideally, each PSU collects individuals of each age within the defined age range, resulting in an equal distribution of age. To address these limitations, when comparing the data from the studies or pooling the data, the influence of those factors (age, sex, sampling years, season, smoking and educational level) on the concentration levels will be investigated using linear regression models per dataset. Significantly influencing factors will be taken into account in the further statistical analysis, by including these variables in the models.

Comparison of results from the individual contributing studies (PSUs) should be performed with caution as not all studies are representative on the national level and therefore results may not in all cases be generalized to the particular country in which they were collected.

Finally, combining HBM studies in Europe also allowed HBM experts from different European countries to exchange information on best practices. The resulting network is of great value to support capacity building and future harmonization of HBM studies in Europe such as under the new partnership on chemical risk assessment (PARC).

### 4.2. Future Plans within HBM4EU

With the completion of the HBM4EU-aligned studies, quality-assured human biomonitoring data will be available to support policies and the upcoming chemicals strategy for sustainability and for a non-toxic environment [53]. The results will provide a baseline for “current” internal exposure levels of European citizens. All metadata and aggregated results will become publicly available. HBM data will be used for deriving European values of internal exposure to environmental pollutants and building policy relevant to Human Biomonitoring Indicators of Chemical Exposure in the European Population including indicators of health risk. Health risk indicators may express the fraction of the study population exceeding established health-based guidance values such as BE values [54] or HBM-GV derived in the HBM4EU project [55,56,57,58] and the extent of exceeding these guidance values (EE) [59]. Geographical differences in exposure distributions and exposure determinants will be analyzed. In a subset of the HBM4EU-aligned studies of children and teenagers, specific effect biomarker analysis will be implemented to gain knowledge on exposure–effect pathways.

In a later phase, additional biomarkers will be measured in the samples of the HBM4EU-aligned studies, i.e., pesticides (glyphosate, AMPA, TCPy, and pyrethroid metabolites: cis-DBCA, cis-DCCA, trans-DCCA, 3-PBA, ClF3CA) and acrylamide (GAMA, AAMA) in children and adults, arsenic species (As(III), As(V), arsenobetaïne, DMA, MMA and total As) in teenagers, UV-filters (Benzophenones: BP-1, BP-2, BP-3, and BP-7) in teenagers and adults and mycotoxins (total DON) in adults. These additional biomarkers are included in the second list of HBM4EU priority substances.

Two more studies joined the HBM4EU-aligned studies for these additional analyses: ORGANIKO (Organic diet and children’s health; Limassol, Cyprus) and RAV-MABAT (The National Health and Nutrition Survey 2015, Israel) in children and adults. Table 6 provides an overview of the basic characteristics of ORGANIKO and RAV MABAT. Further information on questionnaires adopted in ORGANIKO and RAV MABAT is available in Appendix A.

## 5. Conclusions

Overall, the adopted approach results in a European population sample of 3137 children (6–12 years), 2950 teenagers (12–18 years), and 3522 adults (20–39 years) that is representative of sex and has European wide coverage. In most sampling sites, it was difficult to recruit children and teenagers from lower-educated households and adults attaining lower secondary education resulting in underrepresentation in the HBM4EU sample. It is anticipated that the HBM4EU-aligned studies will facilitate qualified statements regarding the internal exposure of Europe’s children, teenagers, and adults to HBM4EU priority substances, stratified by sex and European region. However, when comparing exposure data from the individual contributing studies (PSUs), caution is needed for the interpretation at the country level as not all studies are nationally representative. In addition, heterogeneity among PSUs in sampling years, age, season of sampling, and number of subjects within an age group should be considered if exposure data between studies are compared. More upfront alignment will improve the potential of the European HBM dataset to extrapolate exposure results to the entire European population. To compare exposure levels between European countries, stricter criteria will need to be applied to achieve an equal age distribution per PSU, a shorter time frame, and representativeness on the national level for the collected subsample per PSU. In practice, these criteria are difficult to achieve within the context of alignment of HBM studies where the studies are only partially funded through European projects, and thus national or regional interests in terms of target population and time frame cannot simply be ignored. Gradually, these points can be improved by streamlining national/regional HBM programs.

With the chemicals strategy for sustainability, the European Parliament called upon the commission to collect HBM data to support chemicals risk assessment and risk management. It underlines the need for a clear commitment to securing funds for HBM and environmental monitoring of impacts and exposure to chemicals in order to improve chemical risk assessment and management, as well as for improved sharing and use of local, regional, national, and EU-level monitoring data between countries, sectors, and institutions in relevant policy areas (e.g., water, chemicals, air, biomonitoring, health) [63]. This will be addressed in PARC (Partnership on Chemical Risk assessment) under Horizon Europe where experiences from the HBM4EU-aligned studies will be considered in the further development of a European HBM platform in support of chemicals risk assessment and management.

## Figures and Tables

**Figure 1 ijerph-19-06787-f001:**
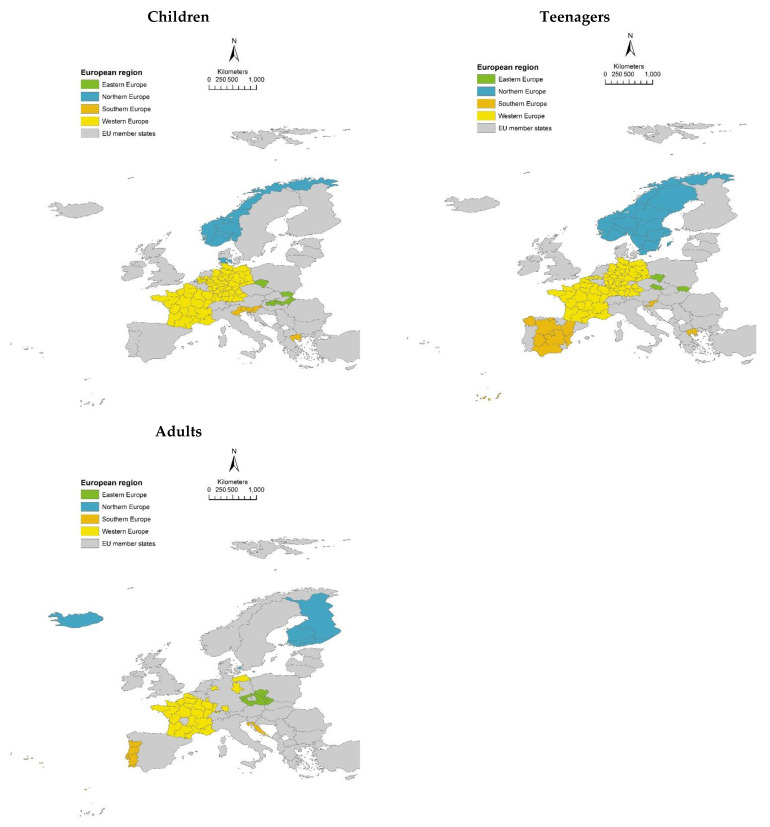
Geographical distribution of children, teenagers and adults’ study populations based on NUTS 2 classification. NUTS = nomenclature of territorial units for statistics developed by EUROSTAT [45]. Upper left = children, upper right = teenagers, bottom left = adults.

**Figure 2 ijerph-19-06787-f002:**
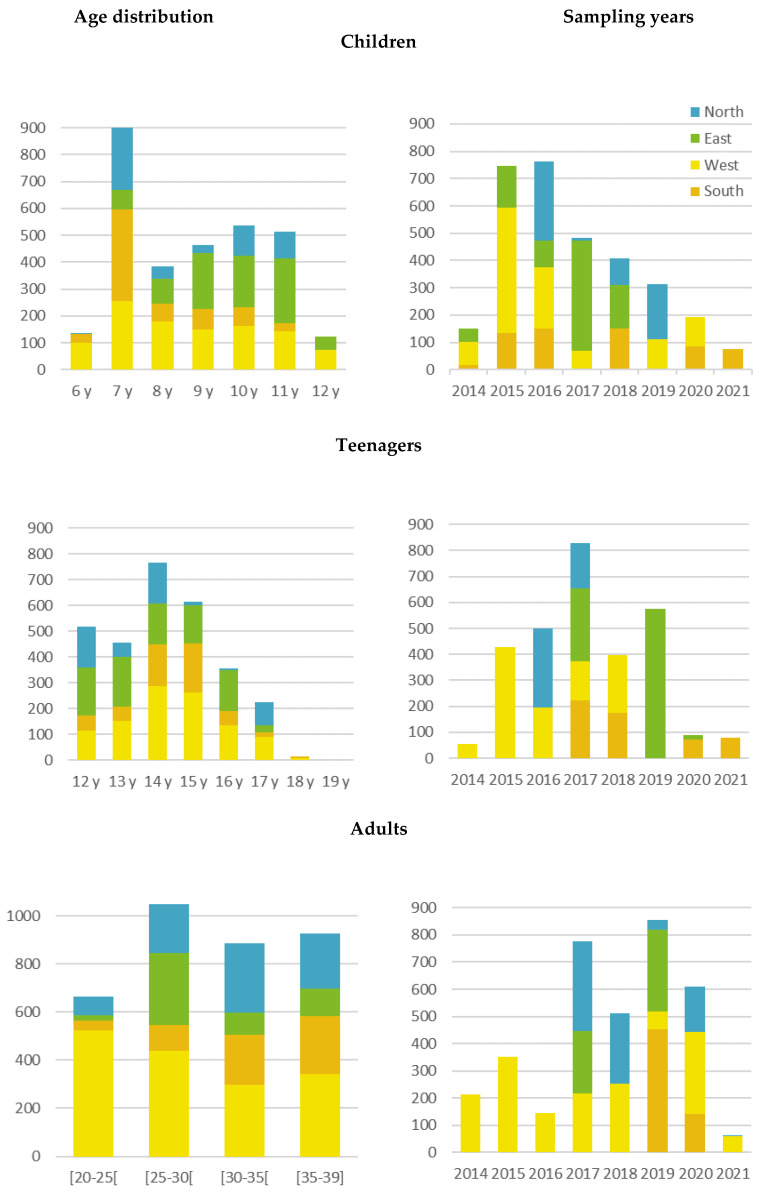
Age distribution and sample collection years covered per European region for children, teenagers and adults.

**Table 1 ijerph-19-06787-t001:** Characteristics of studies contributing to the HBM4EU-aligned studies.

Study Acronym	Location	Level of Representativity	Region	Study Design	Sampling Time Period	Original Age Range	Total Number of Study Subjects	N Selected for HBM4EU	Reference
BEA	Spain	National	-	Cross sectional	10/2017–02/2018	14–16	499	300	[22]
CELSPAC: YA	Czech Republic	Regional	South Moravia	Longitudinal	03/2019–12/2022	28–31	800	300	[23]
CELSPAC:TE	Czech Republic	Regional	South Moravia	Cross sectional	2019–2020	12–17	365	300	-
CPHMINIPUB (parents)/DYMS	Denmark	Regional	The Capital Region of Denmark	Cross sectional	03/2017– 02/2019	20–39	292	292	[24]
CROME	Greece	Regional	Thessaloniki	Cross sectional	07/2020–03/2021	6–18	560	161 children 150 teenagers	-
Diet-HBM	Iceland	National	-	Cross sectional	10/2019–12/2020	20–39	205	205	-
ESB	Germany	Regional	Münster, Greifswald, Halle/Saale and Ulm	Cross sectional	Earliest samples from 1981, ongoing study	20–29	500 per year	700	[25,26,27]
ESTEBAN	France	National	Mainland France	Cross sectional	04/2014– 03/2016	6–74	592 (6–11 y), 512 (12–17 y), 2503 (18–74 y)	543 447 393	[28,29]
FinHealth	Finland	National	Mainland excl. Åland	Cross sectional	01/2017–06/2017	≥25 years	300	300	[30]
FLEHS IV	Belgium	Regional	Flanders	Cross sectional	09/2017–06/2018	14–15	428	300	[31]
GerES V-sub (unweighted)	Germany	National	-	Cross sectional	01/2015–06/2017		2294	300 children 300 teenagers	[32]
HBM survey in adults in Croatia	Croatia	National	-	Cross sectional	11/2019–01/2020	20–39	300	300	-
HBM4EU-study for Switzerland	Switzerland	Regional	Basel	Cross sectional	02/2019– 10/2020	20–39	300	300	-
InAirQ	Hungary	National	-	Cross sectional	11/2017–03/2018	8–11	262	262	[33]
INSEF-ExpoQuim	Portugal	National	-	Cross sectional	05/2019–03/2020	28–39	296	296	-
NAC II	Italy	Regional	Trieste	Longitudinal	08/2014–12/2016	6–8	487	300	[34]
NEB II	Norway	National	-	Longitudinal	2016–2017	7–14	668	300 children 181 teenagers	[35]
OCC	Denmark	Regional	Fynn region	Longitudinal	2018–2019	7	2449	300	[36]
Oriscav-Lux2	Luxembourg	National	-	Cross sectional	06/01/2016–31/01/2018	25–80	1558	210	[37]
PCB cohort/PCB cohort (follow-up)	Slovakia	Regional	Michalovce region	Longitudinal	2014–2017	10–12, 15–17	Original: 415, follow-up: 297	300 children 294 teenagers	[38]
POLAES	Poland	Regional	Lower Silesia	Case-control	09/2017–12/2017	6–11		300 children 281 teenagers 228 adults	-
Riksmaten Adolescents 2016–2017	Sweden	National	-	Cross sectional	09/2016–05/2017	10–21	1305	300	[39]
SLO CRP	Slovenia	Regional	Mura region	Cross sectional	01/2018–06/2018	7–10, 12-15	246	149 (children) 97 (teenagers)	[11]
SPECIMEn-NL	The Netherlands	Regional	Central-East	Cross sectional	01/2020–03/2020	6–11	102	102	-
3 × G	Belgium	Regional	Dessel, Mol, Retie	Longitudinal	01/2019–06/2021	6–8	212	212	-

**Table 4 ijerph-19-06787-t004:** Characteristics of sampled population teenagers by European region and in total.

Characteristics	Northern EU	Eastern EU	Southern EU	Western EU	EU Total	EU Reference
No. of participants	481	875	547	1047	2950	
Age (years)						
Median (p25–p75)	14 (12–14)	14 (13–15)	14 (14–15)	14 (13–15)	14 (13–15)	
Min-max	12–17	12–17	12–18	12–18	12–18	
Missing	0 (0%)	0 (0%)	4 (0.7%)	0 (0%)	4 (0.1%)	
Sampling period (years)						
Median (p25–p75)	2016 (2016–2017)	2019 (2017–2019)	2018 (2017–2020)	2016 (2015–2017)	2017 (2016–2019)	
Min-max	2016–2017	2017–2020	2017–2021	2014–2018	2014–2021	
Sex N (%)						
Girl	254 (52.8%)	423 (48.3%)	274 (50.1%)	541 (51.7%)	1492 (50.6%)	48.38%
Boy	227 (47.2%)	452 (51.7%)	273 (49.9%)	506 (48.3%)	1458 (49.4%)	51.62%
Residential degree of urbanization N (%)						
Cities	155 (32.2%)	380 (43.4%)	337 (61.6%)	240 (22.9%)	1112 (37.7%)	41.70%
Towns/suburbs	219 (45.5%)	187 (21.4%)	65 (11.9%)	450 (43.0%)	921 (31.2%)	31.00%
Rural area	106 (22.0%)	283 (32.3%)	145 (26.5%)	357 (34.1%)	891 (30.2%)	27.30%
Missing	1 (0.2%)	25 (2.9%)	0 (0%)	0 (0%)	26 (0.9%)	
Educational level of the household N (%)						
ISCED 0–2	23 (4.8%)	20 (2.3%)	69 (12.6%)	68 (6.5%)	180 (6.1%)	26.00%
ISCED 3–4	114 (23.7%)	420 (48.0%)	151 (27.6%)	404 (38.6%)	1089 (36.9%)	46.10%
ISCED ≥5	329 (68.4%)	396 (45.3%)	307 (56.1%)	575 (54.9%)	1607 (54.5%)	27.90%
Missing	15 (3.1%)	39 (4.5%)	20 (3.7%)	0 (0%)	74 (2.5%)	

EU reference population based on EUROSTAT tables: residential degree of urbanization based on EU-28, 2017; educational level of the household based on EU-28, 2017, age 15–64 years; sex based on EU-28, 2017, age 15–19 years.

**Table 5 ijerph-19-06787-t005:** Characteristics of sampled population adults by European region and in total.

Characteristics	Northern EU	Eastern EU	Southern EU	Western EU	Total	Reference EU
No. of participants	795	528	596	1603	3522	
Age (years)						
Median (p25–p75)	31 (28–35)	28 (27–34)	33 (30–37)	27 (24–33)	30 (26–35)	
Min-max	20–39	20–39	20–39	20–39	20–39	
Sampling period (year)						
Median (p25–p75)	2018 (2017–2019)	2019 (2017–2019)	2019 (2019–2019)	2017 (2015–2019)	2018 (2017–2019)	
Min-max	2017–2021	2017–2019	2019–2020	2014–2021	2014–2021	
Sex N (%)						
Women	393 (49.4%)	313 (59.3%)	330 (55.4%)	818 (51.0%)	1854 (52.6%)	49.35%
Men	402 (50.6%)	215 (40.7%)	266 (44.6%)	785 (49.0%)	1668 (47.4%)	50.65%
Residential degree of urbanization N (%)						
Cities	645 (81.1%)	443 (83.9%)	254 (42.6%)	1191 (74.3%)	2533 (71.9%)	41.80%
Towns/suburbs	86 (10.8%)	30 (5.7%)	151 (25.3%)	233 (14.5%)	500 (14.2%)	38.20%
Rural area	59 (7.4%)	43 (8.1%)	191 (32.0%)	179 (11.2%)	472 (13.4%)	20.00%
Missing	5 (0.6%)	12 (2.3%)	0 (0%)	0 (0%)	17 (0.5%)	
Educational level of participant N (%)						
ISCED 0–2	153 (19.2%)	2 (0.4%)	60 (10.1%)	22 (1.4%)	237 (6.7%)	16.20%
ISCED 3–4	222 (27.9%)	165 (31.2%)	215 (36.1%)	254 (15.8%)	856 (24.3%)	44.80%
ISCED ≥5	407 (51.2%)	360 (68.2%)	321 (53.9%)	1323 (82.5%)	2411 (68.5%)	39%
Missing	13 (1.6%)	1 (0.2%)	0 (0%)	4 (0.2%)	18 (0.5%)	
Smoking behavior N (%)						
Non smoker	679 (85.4%)	459 (86.9%)	425 (71.3%)	1303 (81.3%)	2866 (81.4%)	
Smoker	105 (13.2%)	66 (12.5%)	163 (27.3%)	283 (17.7%)	617 (17.5%)	22.9%
Missing	11 (1.4%)	3 (0.6%)	8 (1.3%)	17 (1.1%)	39 (1.1%)	

EU reference population based on EUROSTAT tables: residential degree of urbanization based on EU-27, 2017, age 20–39 years; educational level of the participant based on EU-28, 2017, age 25–34 years; sex based on EU-28, 2017, age 20–39 years; smokers based on EU-26, 2014, age 18–44 years.

**Table 6 ijerph-19-06787-t006:** Characteristics of ORGANIKO and RAV MABAT study.

Study Acronym	Location	Level of Representativity	Region	Study Design	Sampling Time Period	Original Age Range	Total Number of Study Subjects	N Selected for HBM4EU	References
ORGANIKO	Cyprus	Regional	Limassol	Cross-over	01/2017–04/2017	10–12	191	166	[60]
RAV-MABAT	Israel	National		Cross-sectional	2015–2016	4–11 year and 18–64 years	Children 103 Adults 194	Children 103 Adults 194	[61,62]

## Data Availability

See Section 3.4. Data Accessibility.

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
