# Peer review of "Harmonization of Human Biomonitoring Studies in Europe: Characteristics of the HBM4EU-Aligned Studies Participants"

_ijerph, 2022, doi:10.3390/ijerph19116787_

Round 1

Reviewer 1 Report

The authors undertook considerable efforts to make the text meaningful and readable.

This manuscript is a resubmission of an earlier submission. The following is a list of the peer review reports and author responses from that submission.

Round 1

Reviewer 1 Report

I read and re-read the paper with a mixed feeling: huge amount of work and no concrete results! I guess, the authors need to published the achievements on the grant money received (The HBM4EU co-financed under Horizon 2020, grant agreement No 733032 plus additiional co-financing by NIRAS, STORA and MONA foundations in participating countries).  According to the title "Harmonisation of human biomonitoring studies in Europe: characteristics of the HBM4EU aligned studies participants", one may expect the RESULTS of harmonization. But where are they? Detailed description of the sampling campaign is impressive indeed: in total, biological samples from 3137 children aged 6-12 years, were collected for  the analysis of biomarkers for phthalates, HEXAMOLL® DINCH and flame retardants. Samples  "from 2950 teenagers aged 12-18 years were collected for the analysis of biomarkers for phthalates,  Hexamoll® DINCH and per- and polyfluoroalkyl substances (PFAS), and samples from 3522 adults aged 20-39 years were collected for the analysis of cadmium, bisphenols and metabolites of poly- aromatic hydrocarbons (PAHs)". What kind of comparison can be underatken if different analytes are involved?

Phthalates,  Hexamoll® DINCH and per- and polyfluoroalkyl substances (PFAS) and flame retardants are in one group; phthalates,  Hexamoll® DINCH and per- and polyfluoroalkyl substances (PFAS) in the second group and cadmium, bisphenols and metabolites of poly- aromatic hydrocarbons (PAHs) in the third group.

The authors state themselves (line 702-707): "As a result, data on exposure to phthalates 703 & HEXAMOLL® DINCH is available for 2880 children and 2799 teenagers. Data on BFRs 704 and OPFRs is available for 711 and 1770 children respectively. Data on PFASs exposure is 705 available for 1957 teenagers and data on biomarkers for bisphenols, Cd and PAHs are 706 available for 2756, 2510 and 2609 adults, respectively".

And further on (lines 776-779): "...This collaborative effort resulted in a total of 9609 analysed samples: 3137 in children,  2950 in teenagers and 3522 in adults with results that are controlled for analytical quality  and comparability. Although the quality and comparability of analytical results is guaranteed through the extensive QA/QC scheme organized as part of HBM4EU [15], continued effort will be needed to improve the analytical capacity in Europe". Well, why to publish this descriptive summary of the work BEFORE the improvement of the analytical capacity in Europe?

To show that I carefully examined the manuscript, I ask the authors of the editors: who has insirted remarks "Reference not found" in lines 498, 499, 513, 530-532, 550, 553, 651, 672, 700, 741?  In some cases references are not necessary. I also noted a misprint in word "analysed" (Tables S13 - S15), z should be instead of s.  

In my opinion, this compilation of the sampling and analyses (without their demonstartion!) in participating countries should be publised in some Pre-print-Report of the job done. In the given form the paper has no scientific importance and can't be useful for the readers.  Recommendations to improve approach, i.e. to overcome "...major limitation of the study: the use of different questionnaires implying the need for post-harmonisation of the variables...  Upfront alignment  of questionnaires in future surveys will increase the comparability and use of the European pooled dataset. Pooling data from different studies increases the sample size and statistical power to study exposure-effect associations..." say that the planning of the study was not well enough thought over...

Reviewer 2 Report

The manuscript written by Gilles et al.: "Harmonization of human biomonitoring studies in Europe: characteristics of the HBM4EU aligned studies participants" provides a summary of the HBM4EU efforts thus far. The work is certainly of interest to the broader community that reads IJERPH.  

Overall the manuscript is quite thorough in documenting characteristics of participants, geographical location, etc. and there are no major scientific concerns for the work that was conducted. I do have some other comments that I believe would help the manuscript, however. 

The abstract, line 62, suggest changing "is becoming" to "has become". 

Adding the specific aims/objectives of the work described in the manuscript to the introduction would significantly improve the readability. One must get to the results/discussion section before it is suggested that aligning the HBM4EU population characteristics with those in the broader whole of Europe on the bases of gender, education, etc. was a goal. 

The background could be enhanced by a brief few sentences highlighting other global biomonitoring efforts to contextualize HBM4EU.

Suggestion to adapt the Methods section documenting all of the various studies into a table with columns for study name, country of origin, sampling strategy, n in study, and n selected for HBM4EU, and other notes. 

For studies where the n selected for HMB4EU were a subset of the total n, were those selected randomly or selective to achieve the desired population characteristics in HMB4EU?

Section 2.3, line 203 seems to read that children may have siblings and spouses; please reword.

Section 2.5 line 235 notes blood samples were only available for a subset of children/teenagers - why were other children/teenagers included in HBM4EU if no biospecimens are available to do biomonitoring?

A brief description of how the biomonitoring measurements were done (and QA/QC) is needed in section 2; some of this exists in section 3.

A minor point - but the Figures do not lend themselves to be printed out in grayscale. 

The tables throughout the results section are very thorough and informative.

Section 3.5 - for most of the chemicals there is a clear preferred matrix - however, Cd is often measured in blood or urine (and is less than useful in serum/plasma). Some granularity for that in Table 4, or perhaps a footnote if all conform to a specific matrix, would be beneficial. Also, if any chemicals are measured in a secondary matrix, that would be worth noting. 

Line 744-745, not sure I understand. Why would only data below the LOD be imputed for use?

Fix the "Error! Reference source not found" throughout the manuscript.

Reviewer 3 Report

The manuscript describes the organisation of the first HBM4EU aligned studies implemented to obtain comparable human biomonitoring (HBM) data of European citizens to monitor their internal exposure to environmental chemicals.  The data resulting from these measurements are not provided. The authors discuss the strengths and limitations of the HBM obtained data, which is necessary for informed decision-making on environment and health policies.

The study is relevant to the field as it addresses a critical issue for human biomonitoring: the difficulties of assembling a large HM database composed of robust and comparable data obtained by different studies. Although the manuscript has the potential to be relevant for the field, the information provided is not always clear and presented in a well-structured manner. The description of the PSUs (Subsections of Section2) should be consistent, always providing the same type of information, namely the participants’ contribution to the studies, the sample size and the date of sample collection. The HBM4EU dataset has limited information on specific ages, and perhaps it would be worth mentioning this as a limitation of the study (it was omitted in the discussion). There are also temporal (e.g. seasonality) and spatial (e.g. predominance of urban settings) differences between studies that can hinder comparisons between countries/geographical regions (another issue excluded from the discussion). There was an error with the numbering of figures and tables in the body of the manuscript (references to figures and tables in the text are illegible), which made the review more difficult than anticipated. Section 3.4 of the Results would fit better in section 2 on Methods, as it merely describes the biological specimens collected from the different PSUs. Similarly, Section 3.5 - Exposure assessment contains only an exhaustive list of the exposure biomarkers measured by the various PSUs and the number of biological samples available for the different chemical substances. Therefore, it would make more sense to include it in the Methods section. Overall, the Results section doesn’t present results. It is just a description of the different methods used by the various PSUs. In the Discussion, the authors discuss several limitations but don’t point out solutions or inform on the strategy that will be used to overcome these limitations. If I understood it well, the experimental design is already published elsewhere (please see line 113), and no data are provided by the authors that allow for assessing the reproducibility of the results. In general, the tables are appropriate and easy to interpret and understand. The references cited are relevant and of good quality. To the best of my knowledge, no similar review was published recently, but there is information on strategies that were omitted and referred to as published elsewhere (e.g. lines 113 and 669) that could be of interest here. Written English needs to be improved.

Detailed comments are listed hereunder.

Lines 132-135: I didn’t completely understand the meaning of “comparing the actual study design with the ambition to obtain quality controlled and comparable HBM data ….”. Is there a study design that is not actual but conceptual? Can the authors please rephrase or elaborate a bit?

Lines 148-151: Is the stipulated timeframe 2014 –2020? If so, I suggest replacing the 2nd criterion with “(ii) studies earlier to the HBM4EU project but with sampling within the 2014-2020 timeframe”.

Lines 195-196: The sentence describes well the contribution given by the participants to the study (samples, questionnaires, and others). In my opinion, this stands for relevant information about each study. However, this information is not provided for the wide majority of the PSUs, and therefore, their characterization is less informative. In my opinion, the specific contribution of the participants must be well described in all studies.

Lines 233-235: If only 107 (55+52) blood samples were obtained from the CROME study, why were 311 participants included in the HBM4EU aligned studies? Isn’t the existence of a biological sample per participant a requisite?

Lines 243: Please check whether the age range is 20 -39 or 20-29 (line 239). It looks inconsistent given that all Diet-HBM participants were recruited from the Icelandic National Dietary Survey (ages ranging from 20-29).

Lines 313-314: Please elaborate on the GerES V-sub(unweighted). I don’t understand what it stands for.

Line 344: Did the participants provide questionnaire data? If so, and considering that the participants are children, how was the data obtained?

Lines 362-363: In my opinion, this sentence requires demands further information. What is the sampling related questionnaire? And the general one? Were they self-administered or obtained through informal interviews?

Lines 386-381: Information on the date of sampling is not provided by the authors. Kindly add it to the text.

Line 404. I suggest replacing the word “wave” with “project”.

Lines 425-426: I suggest including the value of N used in the HBM4EU aligned studies sample for each age group.

Lines 430-431: In my opinion, it would be important to know which biological samples were collected for HBM purposes. I suggest adding the information.

Lines 458-460: Is it possible to provide the total number of samples (N)?

Lines 515-517: Regarding the “individuals from the other studies”, are these individuals from the same age group (children)?

Lines 536: 3.9% of ALL the children sampled were 12 years of age?

Lines 536-548: I’m not sure whether wouldn’t be pertinent to note this as a potential limitation of the study. I would very much like to know the author’s opinion.

Lines 549-554: I think this is a limitation of the study, and it should be indicated as such.

Line 555: I suggest rephrasing it as “Also, the sampling dates differed among PSUs.”, or something similar.

Lines 558-559: I suggest replacing “sampling years covered” with “sampling period”. Does this information concern the HBM4EU aligned studies? It’s not clear from the sentence.

Lines 560-561: Isn’t this another limitation of the study in terms of the representativeness of the study population? Shouldn’t it be indicated as such? I have a similar comment for lines 561-565.

Line 566: I suggest replacing “the sampling years differed per study” with “the sampling dates varied between studies” or something similar. Please revise this issue throughout the text.

Lines 574-576: Therefore, differences in specific ages and sampling dates are limitations of the study that must be overcome through modelling techniques. Isn't it important to highlight this?

Lines 623-624: It would be interesting to know if the underrepresentation of the lower level of education is related to the underrepresentation of the rural areas.

Lines 711-737: I think that the list of metabolites can be deleted since it is presented in the Supplementary Information. A long list of chemical substances in the main text doesn't grasp the readers' attention.

Lines 758-759: The authors identified a limitation of the study (underrepresentation of one level of education) but didn’t make an effort to look for the causes. If the number of uneducated people in Europe is small, the group will always be underrepresented. However, were there biases in the recruitment methods that may have resulted in the underrepresentation? Only a few % of the participants are from rural areas. Are these variables correlated? Can this be a possible explanation for the underrepresentation? This type of information is important for the planning of future studies.

Lines 761-764: I don’t understand the sentence. Perhaps the authors can rephrase it.

Lines 795-796: Considering that the authors know the datasets resulting from the questionnaires used by the different PSUs, they already know the limitations they are facing. Therefore, it seems that it would have been possible to discuss and define the list of variables mentioned here. It would have increased the relevance of the paper.

Lines 817-818: If the HBM4EU has allowed European countries to exchange information on best practices, why were not those practices identified here? Knowing the lessons learned and the solutions found has undoubtedly broad interest and would have increased the value of the study.